# The Pilot Study on the Effect of a Compassion-Based Program on Gifted Junior High School Students' Emotional Styles, Self-Compassion, Empathy, and Well-Being

**Min-Ying Tsai**

Department of Special Education, National Pingtung University, Pingtung City 900391, Taiwan;
minying@mail.nptu.edu.tw

**Abstract:** This study investigated the impact of compassion-based programs on gifted students' emotional style, self-compassion, empathy, and well-being. The study conducted a quasi-experimental study of 30 academically gifted junior high school students in the eighth grade in Taiwan. Seventeen students were in the experimental group, and 13 were in the control group. The study adopted an emotional style scale, a self-compassion scale, an empathy scale, and a well-being scale. Covariance analysis was used to analyze the data. The results found that the students in the experimental group scored significantly higher than the students in the control group on some subscales and total scores of four tests in part. Students also learned more about themselves, identified and adjusted cognitive patterns, communicative skills, and so on, based on their feedback. The results provide some suggestions about future curriculum design, instruction, and counseling based on positive psychology.

**Keywords:** emotional style; empathy; self-compassion; well-being; compassion-based intervention program





## 1. Introduction

Researchers have studied the relationship between psychological well-being and giftedness using a large amount of data for 50 years, finding two opposing viewpoints [1–11]. The first primary hypothesis posits that gifted individuals are more adaptable [5]. Due to their high cognitive abilities and self-reflection, they are more adept at resolving conflicts, overcoming struggles, and developing asynchrony, which contributes to their psychosocial well-being. The second hypothesis suggests that gifted people may be more vulnerable to interpersonal conflicts, impairing their adaptability and well-being, particularly during adolescence and adulthood [5]. From this view, giftedness can lead to further development and cause positive or negative effects, such as achievement, interpersonal conflicts, maladaptation, etc. The psychological functioning of gifted students can be conceptualized from a traditional lens focused on identifying and remedying within-person problems or from a modern lens that takes a more holistic view of individuals as having personal strengths and environmental resources [12]. The first hypothesis originated from the modern lens, which emphasizes their strength and potential development and consultation, and the second hypothesis was based on the traditional lens, which focused on gifted students' maladjustment. Gifted teenagers experience the double stress of their inner physiological variation and outer academic requirements, and they also encounter inner crises of self-identity in adolescence. Thus, the study adopted the second hypothesis and developed an appropriate intervention experimental curriculum for gifted junior high school students.

Gifted students tend to exhibit higher physical, psychomotor, emotional, spiritual, intellectual, and social development levels than typical students, based on Dabrowski's theory [13]. Most studies indicate that the social and emotional issues of gifted students include the following: emotional intensity, perfectionism, deep concern, high sensitivity,

overexcitability, sensitivity to social justice, high self-criticism, difficulty in adapting to new environments, and the lack of social and communication skills [14–17]. These issues can emerge due to asynchronous development, high expectations from various parties, difficulty adapting to social interaction, and so on [16,18]. Perfectionism in gifted students is also a concern, as they may be under pressure or become angry if they cannot perform as perfectly as expected [16]. Based on the above statement, the study hypothesized that gifted students may have potential psychosocial issues. They must cultivate positive adjustment abilities to cope with internal or external pressures. Parents and teachers in Taiwan expect them to behave well and get excellent grades. Gifted students need to learn to recover from stress or a bad state and be kind and understanding toward themselves in pain or failure rather than strictly self-critical. When they adjust their negative emotions, they can maintain stable emotions and behave well. Therefore, the study tried to design a compassion-based curriculum for gifted junior high school students to develop positive adjustments.

Social and emotional difficulties may arise because of gifted students' asynchronous development, exceptional abilities, and notable talents, especially during the teen years [19,20]. Relative literature suggests a variety of areas and conditions for which they may require special assistance, including depression [21,22], social isolation [23], emotional intensity and heightened sensitivity [21,24], feeling different from others [25,26], perfectionism [21,27], social skills deficits and peer relationship issues [22,28,29], and stress management problems [20]. Their environment and culture significantly influence gifted adolescents with higher social and emotional needs. These issues, alone or in combination, can complicate other problems, such as parental separation or divorce, an unstable home life, personality conflicts, grief, behavioral issues, or motivational deficits [21]. Gifted teenagers' moods are easily affected by the outside environment during puberty. Difficulty with emotional regulation or negative perfectionism also increase their vulnerability to psychological problems [30–33]. It is necessary to emphasize their social and emotional needs and design suitable, effective curricula for them, such as self-discipline, compassion, impulse control, pressure management, etc.

When thinking about compassion, people often think about having compassion for others. However, many traditions stress the importance of self-compassion in fostering the emotional resources necessary to give compassion to others, such as in Buddhism [34]. When someone makes mistakes or encounters difficulties, they can show compassion and comfort themselves to recover positive emotions and not blame themselves or feel guilty. Gifted students with perfectionism are often self-critical and feel guilty if they make mistakes or misbehave. If they do not comfort or give compassion to themselves, they might be anxious because they feel like they are not good enough. High school gifted students' self-compassion is positively related to psychological well-being and negatively associated with academic burnout [35]. In addition, the perfectionism of high school gifted students is significantly negatively associated with psychological well-being and significantly positively related to academic burnout [35]. Positive emotions and grit can be a source of overcoming difficulties [36,37]. People who do good things based on their abilities and strengths will produce a sense of well-being and meaningful lives [38]. Junior high school gifted students in Taiwan retained a positive attitude toward learning development, life adaptation, and future career aspects [39]. The study aimed to explore the effect of the compassion-based curriculum design on the emotional style, self-compassion, empathy, and well-being of gifted students in junior high school.

"Social and Emotional Aspects of Learning" (SEAL) is a framework for delivering social and emotional skills to improve students' mental health and behavior. SEAL is a school-based approach to developing children's social and emotional skills, including six topics and five SEAL skills [40]. The National Strategies first evaluated the effectiveness of social and emotional development in raising school achievement in a Behavior and Attendance Pilot in 2003 [41]. Reported benefits of the SEAL approach include improved behavior, attendance, attainment, ways to resolve school improvement, staff development,

leadership, and family and community relations issues [42]. SEAL can help teenagers understand themselves and others' emotional needs and express and enjoy developing their interpersonal relationships. Teachers can involve emotional and social skills to ensure that teenagers maintain the emotional and social health necessary to continue learning.

There is a large body of empirical evidence about the importance of social and emotional skills for successfully navigating one's life [42–45]. The result of a meta-analysis of 213 school-based SEAL programs suggested that they did positively contribute to the healthy development of children, including mental, emotional, and behavioral health, leading to the suggestion that the SEAL approach be incorporated into standard educational practice [44,45]. Based on Goleman's (2004) Emotional Intelligence Inventory, the five SEAL skills include self-awareness, managing feelings (self-regulation), motivation, empathy, and social skills, all of which are structured around an annual cycle of topics designed by the National Strategies that reflect the school experience of children [42,46]. The SEAL program was shown to affect students' healthy development effectively. Social and emotional skills not only influence life outcomes directly (for example, good social competence helps people successfully negotiate job interviews), but they also have persistent and cumulative effects on other attributes, including cognitive skills [43]. Similar studies showed that social and emotional learning skills could decrease problems related to mental health, perfect marriage ideals, and physical health [47]. The study referenced the five core skills in social and emotional learning curricula to construct the curriculum, as seen in Table 1.

**Table 1.** The theme and content of the social and emotional learning aspects.

| Theme | Content | Theory and Strategy |
|---|---|---|
| Self-awareness | <ul><li>The fundamentals of social and emotional learning.</li><li>Helping teenagers accept themselves and constructively interact with others.</li></ul> | <ul><li>Erikson's identity and role confusion stage</li><li>Holland's vocational personality typology theory</li><li>Johari Window</li></ul> |
| Emotional identification | <ul><li>Helping teenagers perceive and analyze their emotions and release negative emotions appropriately.</li></ul> | <ul><li>Examine teenagers' origin of negative emotions, such as inner factor (hormonal stimulation) or external factor (expectation stress)</li></ul> |
| Interpersonal relations | <ul><li>Excessively affect our mental and physical health and life quality.</li><li>Developing and maintaining good interpersonal relations were the essence of their social and emotional health.</li></ul> | <ul><li>Bronfenbrenner's ecological systems theory</li><li>Analysis of the concept of interpersonal influence with the Bull's Eye</li></ul> |
| Emotional and pressure management | <ul><li>It is an important skill to deal with physiological stress, mental challenges from seeking peer acceptance, solving complicated relations, and so on.</li></ul> | <ul><li>Cognitive distortions</li><li>A-B-C model of emotions in Ellis' Rational-Emotional Therapy</li></ul> |
| Communicative skills | <ul><li>There is a high risk of mental health problems in young people because of excessively aggressive or passive social interactions.</li></ul> | <ul><li>Individuals have four primary responses when confronted with threats or in challenging social situations: aggressive, cynical, indifferent, and confident.</li></ul> |

Concerning the experimental research of the social and emotional program in Taiwan, the gifted junior high school students' leadership skills and emotional intelligence are enhanced through the adventure education program [48]. By participating in service-learning programs, gifted students can serve the community, care for the disadvantaged, and improve their interpersonal and emotional management skills [49]. Leadership training

or service-learning benefited students' leadership and interpersonal skills, compassion, and emotional intelligence. University bright underachievers found and performed their strengths to practice tasks and gradually understood the characteristics of their backgrounds reflected on their self-evaluation; then, they showed confidence and motivation to participate and became aware of their efforts to achieve the goals of the task-oriented plans [50]. Students actively participated in the task-oriented plans, which helped them understand their weaknesses and strengths and improve their social and emotional skills.

A junior high school gifted underachiever had a positive transformation of self-understanding, reading habits, and methods through bibliotherapy after one year of consultation [51]. After the termination of the cognitive-behavior approach of bibliotherapy, the prosocial behaviors of the fourth-grade girl increased, such as caring about others, actively comforting people, sharing desserts with classmates, helping others, and decreasing anger [52]. Using blogs in independent study significantly increased gifted students' abilities to self-direct, learn, be motivated, think deeply, and cooperate with peers [53]. By using metaphor stories, inquiry, and practice, fourth-grade students more effectively solved personal problems [54]. Seventh-grade students improved their character and attitudes by reading newspapers and discussing the content [55]. The film-guided teaching with group discussion can effectively improve fourth-grade students' performance and learning motivation in science class [56]. Bibliotherapy, multimedia (e.g., blogs and videos), group discussions, and practice benefit students' character, attitudes, and performance [56].

The positive psychology curriculum significantly increased the positive character and emotions as well as emotional stability of gifted students [44,57,58]. Through the Co-cognitive Traits Intervention program, the analytical thinking, positive motivation, creative thinking, and sensitive, emotional expression of primary gifted students were enhanced and further affected their gifted behaviors [58]. During the Thriving Learning Communities program, fifth- and sixth-grade students discovered and understood their unique constellations of character strengths and improved their social-emotional competencies by cultivating personal character strengths [44]. Two gifted high school students shared that their self-reflection and self-transformation attitudes improved, they learned about self-understanding, and they kept a cheerful mood to maintain good interpersonal relations after the "Ten Tastes of the Well-being program" based on the Mindsight concept [59]. As such, this study listed systematically influential teaching factors to design a curriculum that included self-awareness, self-compassion, positive emotions and traits, empathy, and so on. Regarding instructional strategies, we adopted video discussion to inspire their learning motivation, explain directly, use cognitive modeling, and communicate empathetically.

## 2. Materials and Methods

The study adopted a non-randomized control group, pretest-posttest quasi-experimental design. The dependent variables were the emotional style, self-compassion, empathy, and well-being scale. The independent variable was the compassion-based intervention curriculum. The study used pretest scores as covariates and posttest scores as dependent variables based on the analysis of covariance to decrease the effect of the confounding variables or the control variables. The experimental design is shown in Table 2.

**Table 2.** Experimental design.

| Group | Pre-Test | Experiment | Post-Test |
|---|---|---|---|
| experimental group | $O_1$ | $X$ | $O_2$ |
| control group | $O_3$ | | $O_4$ |

$X$: The compassion-based intervention curriculum; $O_1$, $O_3$: The pre-test of the emotional style, self-compassion, empathy, and well-being scale; $O_2$, $O_4$: The post-test of the emotional style, self-compassion, empathy, and well-being scale.

*2.1. Research Subjects*

The study adopted purposive sampling to select one gifted resource classroom in a junior high school in Kaohsiung. All students passed the gifted identification test by the Education Bureau of Kaohsiung City Government and were identified as academically talented students in the study. Seventeen gifted students in the eighth grade who chose the elective subject of an affective curriculum comprised the experimental group. Thirteen students who did not select the elective subject of the affective curriculum comprised the control group. Since one student in the experimental group did not participate in the gifted resource classroom after the first examination, there were sixteen students in the experimental group. One student in the control group did not come to the gifted resource classroom for an extended period. There were twelve students in the control group who accepted the post-test.

*2.2. Research Instrumentss*

This study adopted four scales: the emotional style, self-compassion, empathy, and well-being scales. Except for the empathy scale, the other three scales were revised by the research team and verified by junior high school teachers and gifted education professors. All scales were pre-tested and suitable for junior high school students in Taiwan to correspond to the cultural adaptation.

2.2.1. The Empathy Scale

The original model of the empathy scale consisted of four subscales, including perspective taking, fantasy, empathic concerns, and personal distress, with seven questions in each subscale. The study adopted the revised sympathy scale developed by Lin (2008) [60]. There were cognitive empathy (perspective-taking) and affective empathy subscales, with six questions in each subscale. A 5-point Likert scale was used in the study [60]. The cognitive, affective, and whole empathy reliability coefficients were 0.81, 0.85, and 0.89, respectively. The factor loading of each question was between 0.63 and 0.81. The total cumulative explanation variance was 55.57%. The reliability and validity were evidenced in Lin's (2008) scale [60].

2.2.2. The Emotional Style Scale

The study referenced the emotional style scale developed by Davidson and Begley (2012) and the translated version of the emotional style scale by Hong (2013) to construct the Chinese version of the emotional style [38,61]. The subscales were social intuition (6 items), situational sensitivity (4 items), resilience (5 items), outlook (4 items), self-awareness (3 items), and attention (3 items). These data were analyzed using confirmatory factor analysis. The composite reliability of each subscale was 0.52~0.82, and the average variance extracted from each subscale was 27.71~43.91%. Constructive validity and reliability were evidenced.

2.2.3. The Self-Compassion Scale

The study referenced the self-compassion scale developed by Neff (2003) to construct the Chinese version of the self-compassion scale [62]. The subscales were self-kindness (5 items), self-judgment (11 items), common humanity (3 items), and mindfulness (6 items), and they were analyzed using confirmatory factor analysis. The composite reliability of each subscale was 0.72~0.89, and the average variance extracted from each subscale was 40.50~55.50%. Constructive validity and reliability were evidenced.

2.2.4. The Adolescent Well-Being Scale

The study referenced three scales to construct the junior high school version of the well-being scale, which combined the concepts of subjective and psychological well-being, including three well-being scales developed by Hsia (2011), Fan (2008), and Huang (2010) [63–65]. The adolescent well-being scale consisted of positive and negative emotions,

mental health, life satisfaction of subjective well-being, self-affirmation, interpersonal relationships, and flow of psychological well-being. The scale was analyzed using confirmatory factor analysis to ensure that each question/number of positive thinking, self-affirmation, interpersonal relationship, life satisfaction, flow, and mental health was 4, 7, 3, 5, 4, and 5. The explanation variances of positive thinking, self-affirmation, interpersonal relationship, life satisfaction, flow, and mental health were 33.93%, 51.47%, 51.00%, 49.09%, 47.07%, and 50.55%, respectively. The composite reliability of each subscale was between 0.72 and 0.88. Constructive validity and reliability were evidenced.

### 2.2.5. Compassion-Based Program

The experimental instruction promoted gifted students' emotional style, self-compassion, empathy, and well-being; it also addressed issues related to development of their emotional and social interpersonal relationships. The curriculum content in the intervention program included six topics: self-exploration, interpersonal network, emotional event, prosocial behavior, interpersonal communication, and prosocial behavior. The curriculum was implemented in a gifted resource classroom once a week, and each lesson had 45 min. The entire experimental program was conducted over a semester for about 18 weeks once a week, as shown in Table 3.

**Table 3.** The theme and goal of the compassion-based program.

| Theme | Goal | Theory |
|---|---|---|
| self-exploration | To help students understand their traits from others' perspectives to appreciate and accept all their traits | • Holland's vocational personality typology theory <br> • the Johari Window |
| interpersonal network | To help students construct a circle of friends who positively impact them, providing a support system that makes them feel connected and valued | • Bronfenbrenner's ecological systems theory <br> • the concept of international influence |
| emotional event | To discover the reasons that trigger emotions, to confirm what emotions are caused by oneself, and to understand how emotions affect one's physiology, language, and behaviors | • the social and emotional aspects of learning |
| physical and mental adjustment | To help students explore their sources of stress, change their irrational cognitive views, and find a relaxing way | • the emotional ABC model <br> • cognitive distortion |
| interpersonal communication | To help students think about whether the communication is too improper and might cause interpersonal conflict, discuss how to solve conflicts, and establish confident communication | • the social and emotional aspects of learning |
| prosocial behavior | To comfort people in bad moods, help them regain self-assurance, and feel understood through the self-compassion concept | • the five responsive methods of empathy <br> • emotional regulation |

### 2.3. Data Analysis

Before and after the curriculum, students completed the four tests and collected their worksheets during the curriculum. This study used analysis of covariance (ANCOVA) to determine whether there was a significant difference between pre-and post-tests for emotional style, self-compassion, empathy, and well-being in the experimental group after the intervention program. When $\eta^2$ is greater than 0.138, it suggests that the experimental treatment and the dependent item are highly correlated based on explanatory power; when $\eta^2$ is more significant than 0.059, it indicates that the experimental treatment and

the dependent item are moderately strong; $\eta$ is more critical than 0.001, representing the experimental treatment and the dependent item. The study was supplemented by the qualitative data of students' study sheets and feedback sheets to support the quantitative data and experimental results. The researcher coded the students' responses from their worksheets designed based on the curriculum content and supplied some additional information about the feedback of the student's opinions about the effect of the curriculum. The code marked students' grades, classes, and personal numbers. For example, the individual number of the eighth-grade student in the second classroom was 23, marked as S20223.

## 3. Results

The results from the experimental and control groups of students regarding emotional style, self-compassion, empathy, and well-being scales are presented below.

### 3.1. The Results from the Experimental and the Control Group of Students Regarding Emotional Style, Self-Compassion, Empathy, and Well-Being Scales

The experimental group of students on the post-test of each subscale scored higher than the pre-test of each subscale (Table 4). The experimental group's post-test scored higher than the control group on six subscales and the total scores. The pre-test of the control group was slightly higher than the post-test in each subscale and the total scores, except for outlook.

**Table 4.** Summary of the pre- and post-tests of the emotional style.

| Subscale | The Experimental Group | | The Control Group | | *F* (*p*) | $\eta^2$ |
|---|---|---|---|---|---|---|
| | Pre-Test (*N* = 17) | Post-Test (*N* = 16) | Pre-Test (*N* = 13) | Post-Test (*N* = 12) | | |
| Attention | 13.94 (2.46) [1] | 14.75 (1.18) | 13.83 (2.89) | 13.67 (2.27) | 2.85 (0.104) | 0.102 |
| Self-awareness | 6.06 (1.88) | 6.72 (0.93) | 6.75 (2.18) | 6.67 (1.231) | 0.05 (0.832) | 0.002 |
| Outlook | 16.88 (4.18) | 18.60 (1.78) | 17.83 (4.17) | 17.88 (2.34) | 0.75 (0.395) | 0.029 |
| Resilience | 18.75 (5.15) | 20.35 (3.18) | 20.58 (3.32) | 20.00 (3.10) | 0.36 (0.557) | 0.014 |
| Sensitivity to context | 21.94 (4.02) | 21.95 (3.27) | 22.08 (3.03) | 20.58 (2.68) | 3.19 (0.086) | 0.113 |
| Social intuition | 25.19 (4.09) | 27.27 (3.89) | 27.92 (5.93) | 26.25 (4.43) | 0.32 (0.575) | 0.013 |
| Total score | 102.75 (12.52) | 109.64 (9.89) | 109.00 (13.93) | 105.04 (9.10) | 3.182 (0.087) | 0.113 |

[1] Note: Means (standard deviation).

The result of the homogeneity of regression coefficients within groups regarding attention, social intuition, self-awareness, outlook, resilience, sensitivity to context subscale, and the total scores did not reach the level of significance. Consistent with the assumption of homogeneity of regression coefficients within the covariate group, the covariate analysis can be continued. According to Levene's test equation, the F value of attention, social intuition, self-awareness, outlook, resilience, sensitivity to context subscale, and total scores did not reach a level of significance. After excluding the influence from previous grades on the later grades, the F values of the practical value of the independent variable on the dependent variables of attention, social intuition, self-awareness, outlook, resilience, sensitivity to context, and emotional style were 2.849 (*p* = 0.104), 0.323 (*p* = 0.575), 0.046 (*p* = 0.832), 0.749 (*p* = 0.395), 0.355 (*p* = 0.557), 3.193 (*p* = 0.086), and 3.182 (*p* = 0.087), respectively, which did not reach a significant level. This indicated that experimental intervention did not promote the subjects' post-test scores of each subscale or the total scores.

### 3.2. The Effect of Self-Compassion from the Intervention Program

In the experimental group, students' post-tests of each subscale were higher than the pre-tests of each subscale (Table 5). The post-tests of the experimental group were higher than those of the control group on four subscales and on total scores. The post-test of the

control group was slightly higher than the pre-test in each subscale and the total scores, except for subscales of self-judgment and common humanity.

**Table 5.** Summary of the pre- and post-test of self-compassion.

| Subscale | The Experimental Group | | The Control Group | | F (p) | $\eta^2$ |
|---|---|---|---|---|---|---|
| | Pre-Test (N = 17) | Post-Test (N = 16) | Pre-Test (N = 13) | Post-Test (N = 12) | | |
| Self-kindness | 16.31 (3.68) [1] | 18.38 (1.59) | 16.67 (3.20) | 17.67 (2.77) | 0.79 (0.384) | 0.030 |
| Self-judgment | 32.75 (6.60) | 33.00 (6.01) | 32.83 (6.83) | 31.50 (8.53) | 0.31 (0.583) | 0.12 |
| Common humanity | 8.69 (2.50) | 11.06 (1.61) | 9.92 (1.51) | 9.75 (1.96) | 5.14 (0.032) * | 0.171 |
| Mindfulness | 18.25 (3.75) | 21.69 (1.96) | 19.33 (3.94) | 20.42 (2.31) | 2.0 (0.170) | 0.074 |
| Total score | 76.00 (12.56) | 84.12 (5.94) | 78.75 (9.30) | 79.33 (8.91) | 4.69 (0.040) * | 0.158 |

[1] Note: Means (standard deviation), * $p < 0.05$.

The result of the homogeneity of regression coefficients within groups about self-kindness, self-judgment, common humanity, mindfulness subscales, and the total scores did not reach a level of significance. Consistent with the assumption of homogeneity of regression coefficients within the covariate group, the covariate analysis can be continued. According to Levene's test equation, the *F* values of self-kindness, self-judgment, common humanity, mindfulness subscales, and the total scores did not reach a significance level. After excluding the influence from previous grades on the later grades, the *F* values of the practical value of the independent variable on the dependent variables of self-kindness, self-judgment, and mindfulness subscales were 0.786 ($p = 0.384$), 0.309 ($p = 0.583$), and 1.996 ($p = 0.170$), respectively, which did not reach a level of significance. This indicates that the subjects' post-test scores of each subscale were not promoted by experimental intervention except for common humanity and the total scores.

The *F* values of the effective value of the independent variable on the dependent variable of common humanity and total scores were 5.139 ($p = 0.032 < 0.05$, $\eta^2 = 0.171$) and 4.685 ($p = 0.040 < 0.05$, $\eta^2 = 0.158$), respectively. The variations of the intervention program to common humanity and total scores were 17.1% and 15.8%, respectively. The statistical powers of common humanity and total scores were 58.7% and 54.8%. The experimental group's common humanity and total scores significantly improved after the intervention program.

*3.3. The Effect of Empathy from the Intervention Program*

In the experimental group, students' post-tests of each subscale were higher than the pre-tests of each subscale (Table 6). The experimental group's post-test was higher than the control group on two subscales and the total scores. The pre-test of the control group was higher than the post-test in each subscale and the total scores. The results of the homogeneity of regression coefficients within groups regarding cognitive empathy, emotional empathy subscales, and the total scores did not reach a significance level. Consistent with the assumption of homogeneity of regression coefficients within the covariate group, the covariate analysis can be continued. According to Levene's test equation, the *F* value of cognitive empathy did not reach a significant level, except for emotional empathy and the total scores. The *F* values of emotional empathy and the total scores reached a considerable level through the Welch/Brown–Forsythe test and rejected the null hypothesis that the groups have equal means.

After excluding the influence from previous grades on the later grades, the *F* value of the practical value of the independent variable on the dependent variable of cognitive empathy was 14.014 ($p = 0.001$, $\eta^2 = 0.359$), and the explained variation of the intervention program to cognitive empathy was 35.9%. The statistical power of cognitive empathy was 94.9%. The ability of students to understand other people's perspectives improved significantly, along with their cognitive empathy. According to the result of the independent *t*-test, the t-value was 2.864 ($p = 0.012$), representing the experimental group of students.

**Table 6.** Summary of the pre- and post-test of empathy.

| Subscale | The Experimental Group | | The Control Group | | $F$ ($p$) | $\eta^2$ |
|---|---|---|---|---|---|---|
| | Pre-test ($N$ = 17) | Post-Test ($N$ = 16) | Pre-Test ($N$ = 13) | Post-Test ($N$ = 12) | | |
| Cognitive empathy | 20.19 (4.32) [1] | 23.13 (2.85) | 19.17 (4.71) | 18.42 (3.90) | 14.01 (0.001) ** | 0.359 |
| Emotional empathy | 20.00 (4.46) | 22.88 (2.13) | 19.08 (5.02) | 18.92 (4.42) | 10.65 (0.003) ** | 0.299 |
| Total score | 40.19 (8.46) | 46.00 (4.68) | 38.25 (8.72) | 37.33 (7.84) | 13.65 (0.001) ** | 0.353 |

[1] Note: Means (standard deviation), ** $p < 0.001$.

After excluding the influence of the previous grades on the later grades, the $F$ value of the practical value of the independent variable on the dependent variable of emotional empathy was 10.652 ($p = 0.003$, $\eta^2 = 0.299$), and the explained variation of the intervention program to cognitive empathy was 29.9%. The statistical power of cognitive empathy was 88%. Students' emotional empathy improved significantly, which led to them feeling more empathy for others.

According to the result of the independent $t$-test, the $t$ value was 3.651 ($p = 0.001$), representing that the experimental group students scored significantly higher than the control group students. After excluding the influence of the previous grades on the later grades, the $F$ value of the practical value of the independent variable on the dependent variable of emotional empathy was 13.653 ($p = 0.001$, $\eta^2 = 0.353$), and the explained variation of the intervention program to cognitive empathy was 35.3%. The statistical power of cognitive empathy was 94.9%. Students' emotional empathy improved significantly, which led to them feeling more empathy for others. Students in the experimental group scored significantly higher on empathy than those in the control group. As a result of the experimental teaching, students are better able to understand the emotional reactions and experiences of others indirectly.

*3.4. The Effect of Well-Being from the Intervention Program*

In the experimental group, the students' post-tests of each subscale were higher than the pre-tests of each subscale, except for flow (Table 7). The experimental group's post-test was higher than the control group on seven subscales and the total scores. The post-test of the control group was higher than the pre-test in each subscale and the total scores, except for self-affirmation and flow.

**Table 7.** Summary of the pre- and post-test of the well-being.

| Subscale | The Experimental Group | | The Control Group | | $F$ ($p$) | $\eta^2$ |
|---|---|---|---|---|---|---|
| | Pre-Test ($N$ = 17) | Post-Test ($N$ = 16) | Pre-Test ($N$ = 13) | Post-Test ($N$ = 12) | | |
| Self-affirmation | 25.25 (5.35) [1] | 27.25 (2.11) | 25.75 (3.67) | 24.75 (3.28) | 7.50 (0.011) * | 0.231 |
| Interpersonal relationship | 11.06 (2.74) | 11.56 (1.55) | 10.58 (1.73) | 10.67 (2.02) | 1.15 (0.295) | 0.044 |
| Mental health | 19.00 (3.35) | 19.56 (1.75) | 18.00 (3.36) | 18.67 (2.15) | 0.94 (0.341) | 0.036 |
| Flow | 16.69 (3.22) | 16.19 (1.71) | 16.00 (3.46) | 15.00 (2.49) | 1.89 (0.182) | 0.070 |
| Positive thinking | 14.56 (3.10) | 15.19 (1.87) | 12.50 (3.06) | 14.33 (2.23) | 0.97 (0.333) | 0.038 |
| Life satisfaction | 18.06 (3.21) | 19.50 (1.83) | 15.83 (4.17) | 17.58 (1.98) | 3.76 (0.64) | 0.131 |
| Total score | 104.63 (18.76) | 109.26 (7.75) | 98.67 (17.13) | 101.00 (10.66) | 4.44 (0.045) * | 0.151 |

[1] Note: Means (standard deviation), * $p < 0.05$.

After excluding the influence of the previous grades on the later grades, the $F$ value of the practical value of the independent variable on the dependent variable of self-affirmation was 7.497 ($p = 0.011$, $\eta^2 = 0.231$), and the explained variation of the intervention program to self-affirmation was 23.1%. The statistical power of self-affirmation was 74.9%. In the

experimental group, self-affirmation scores significantly improved, and students recognized their abilities and appearances. After excluding the influence from previous grades on the later grades, the *F* value of the practical value of the independent variable on the dependent variable of well-being was 4.439 ($p = 0.045$, $\eta^2 = 0.151$), and the explained variation of the intervention program to well-being was 15.1%. The statistical power of self-affirmation was 52.6%. This course can significantly improve students' self-affirmation and well-being, while other variables did not differ significantly.

*3.5. The Feedback from Students' Worksheets*

With regard to self-awareness of emotional style, this is evident in the student feedback in the following statements, such as, "Know me better, all kinds of happy contents" (S20323); "It turns out that I have so many hidden qualities, I know myself better" (S20434); "It seems that I am mysterious, because of how I think about myself" (S21820); "Let me understand my emotions better" (S20409). The students also learned to accept their shortcomings and foster emotional management, conflict reduction, self-esteem, and resilience promotion. This is illustrated through participant responses such as, "I learned how to solve conflicts, and ways to decrease conflict and increase self-esteem" (S21719); "Let me more understand myself and receive my disadvantages" (S20409); and "To know how to manage the emotions effectively" (S20434).

Concerning the viewpoints of self-compassion, students have mentioned that they "Understand positive character" (S20830); " Learn more emotional knowledge and face emotions" (S22214); "More understand self-esteem and self-compassion" (S20830); "Try to change thoughts" (S21820); "Let me learn to release emotions" (S22417); and "Know how to manage emotions effectively" (S20434). In terms of self-confidence and happiness, students mentioned in their feedback that "Increasing self-confidence is great" (S22214) and "Learning positive emotions" (S22218). Speaking of self-criticism, students mentioned that "I am high-critical, and it is normal" (S20830); "Is this criticism enough?" (S22214); "Because we are educated to correct errors when we are small, nobody taught us to acclaim others" (S21719); and "Because asking others to improve can always achieve greater results, at least that is the case with class guides" (S21820).

Based on the feedback about empathy provided by the students, it was evident that they had acquired knowledge on "perceiving others' emotions" (S21033), "helping others overcome difficult times and build mutually beneficial relationships" (S21030), "using empathy to comfort others" (S21719), and "treating others with more empathy" (S22417). Empathic responses can also be found in conversations with others. As an example, students highlighted the importance of empathizing with others through supportive responses (S20409, S20434, S20830, S21033, S21719, S21830), noting that "the injured person can hear varying responses, and there will be varying feelings" (S21719). Additionally, students expressed their opinions on empathy, including the following: "Empathy and sympathy are essential" (S20409); "It is better to put yourself in someone else's shoes than to give alms" (S21033); "Empathy and sympathy sound similar, but make others feel differently" (S21719); "Compassion and empathy help see the good side of things, but not necessarily as a relief; giving makes the world a better place and experiences happiness that we cannot experience ourselves" (S22214). One student mentioned "I am willing to take the initiative to care for others" and "Pursue your interests and help others" to promote confidence and self-esteem (S21030).

Based on the students' feedback about well-being, they better understood the influence of others and how to maintain interpersonal interactions and change their minds. This feedback is illustrated in the following examples: "Let me know how to expand my interpersonal circle so that I can better know how to express my stress" (S20409); "I know more about the influence of people around me, and I have learned many ways to adjust to stress" (S21719); "The personal circle is great; it is great to increase my confidence" (S22214); "Let me know more about interpersonal interactions" (S22417); "Try to change my mind"

(S21820); and "There has been much stress recently so that I can learn the pressure relief method" (S21033).

## 4. Discussion

There was no significant difference between the emotional style subscale and the full scale. The reason might be that the effect of attention, self-awareness, outlook, resilience, social intuition, and situational sensitivity may require long-term effective practice. In addition, the content of the course is rich, but the course on improving social intuition and situational sensitivity needed to be more significant. In addition, a time factor limited the effect of the experimental curriculum. Although students did not have enough time to practice thoroughly in class, they still responded to discussions with quick minds. This result was inconsistent with the results of Bates-Krakoff and McGrath (2017) [44]; the study failed to improve their social-emotional competencies. However, some students mentioned that they were more understanding of themselves, and the content about self-awareness and outlook had a positive impact based on their feedback. Although students did not significantly benefit from the test results, they learned essential messages. The result was similar to Chang and Jhang's (2021) [59]; the study increased their self-understanding.

Regarding the effect of self-compassion, the common humanity and self-compassion of the experimental group were significantly improved after the intervention program. After the intervention program, students were more aware of the concepts of shared humanity and self-compassion based on their feedback. In addition, students also learned about the concepts of emotional management, self-esteem, and self-compassion. This result is similar to the findings of Bates-Krakoff and McGrath (2017) [44] and Chang and Jhang (2021) [59], which improved students' good mood and social-emotional competencies. In short, after acquiring the concepts and strategies of self-compassion, students still need to practice and internalize them for extended periods to gradually reduce self-criticism, maintain optimistic thinking, and comfort and motivate themselves in the face of failures and setbacks. The findings partly matched those of Chang and Pan (2008) [48] and Chen (2014) [49]; the research improved their capacity for emotional regulation, similar to self-compassion and common humanity. Students realized that making errors was normal and they did not have to try to punish themselves by feeling bad or self-critical. The experimental group students did not significantly decrease their self-criticism, for they tended to criticize themselves and others, rarely appraising others. The whole environment in Taiwan makes students see their shortcomings rather than their strengths. Therefore, practicing changing their minds over extended periods is still necessary to decrease their self-criticism and achieve self-kindness.

Regarding empathy's impact, students in the experimental group significantly improved their overall cognitive and emotional empathy. Students also learned about the importance of the empathy response based on their feedback. Through the intervention program, students can observe that different responses to others lead to different outcomes and choose the empathy response. In the course feedback sheet, students mentioned that helping others can improve positive traits by watching videos and discussing empathy. The results were similar to those of Li (2015) [56] and Huang (2006) [52], who stated that film-guided teaching with group discussion can effectively improve performance and learning motivation as well as students actively helping others. However, this study showed that using films can help enhance students' empathy and even lead to helping others' ideas.

Concerning the effect of well-being, students could improve self-affirmation and well-being but not positive thinking, flow experience, physical and mental health, or life satisfaction. This finding was partially consistent with Wang et al. (2006) [51], Chen (2020) Chen [50], and Chang and Jhang (2021) [59], who found a strengthening in gifted students' confidence, self-understanding, and emotional adaptation but not improvements in their positive thinking flow experience, physical and mental health, life satisfaction, or interpersonal interaction (communication). The reason is that the course emphasized self-compassion and emotional management as strategic concepts. During the course,

students were only required to consider the people who affected them most and discuss interpersonal influence and appropriate communication techniques. However, interpersonal relationships take time to change, so they can only be improved after a while. In addition, students needed to check for cognitive distortions in the course content to examine whether their cognitive concepts were biased. It took a long time to practice improving positive thinking, physical and mental health, and life satisfaction. The course content did not incorporate creativity-enhancing components or provide enough time to boost physical or psychological strength. As such, there was no significant difference in interpersonal relationships, positive thinking, physical and mental health, life satisfaction, or flow experiences. However, students can improve their life satisfaction by learning to relieve stress, and understanding self-confidence can help improve self-affirmation.

## 5. Conclusions

This study's results showed no significant difference in attention, outlook, resilience, contextual sensitivity, social intuition, or overall emotional style in the experimental group. However, students' responses to the feedback illustrate improvements in their self-understanding. The students' common humanity and self-compassion scores in the experimental group were significantly improved; the students tended to be self-critical, so there was no significant improvement in self-kindness and mindfulness. The cognitive empathy, emotional empathy, and overall empathy of the students in the experimental group were significantly improved; the students expressed that they had learned how to perceive the emotions of others and used empathic responses to treat others. The self-affirmation and overall well-being of the students in the experimental group were significantly improved. However, there was no significant difference in interpersonal relationships, physical and mental health, positive thinking, life satisfaction, or flow experience. Students in the experimental group learned good communication skills, expanded their interpersonal circle, were influenced by other people, changed their cognition, and so on. However, the intervention's influence on emotional style, interpersonal relationships, positive thinking, life satisfaction, flow experience, and so on still needed a long time to practice and incubate to be better. The program examined in this study can improve gifted students' self-compassion, empathy, and well-being.

This study puts forward three suggestions regarding teaching and counseling for gifted students: First, the course content combined with life experience can guide students to think and solve problems. Based on the class situations in our study, students discussed emotions or conflict situations caused by daily life events. Related to their experiences, students had lively discussions and shared their ideas. Next, when teaching affection courses, we used situations that middle school students may encounter to discuss the emotions these situations may cause and how to solve these problems and to determine if the solution is appropriate when talking about it with each other. The combination of course content and life experience can improve students' learning motivations and help solve real problems in daily life.

Third, class management strategies such as cooperative learning and seating arrangements can promote peer discussions. When students were sitting on the wooden floor in class and arranged in a large circle, it facilitated discussion and improved concentration. As such, teachers can successfully carry out related activities through space planning. For specific topics, group discussions can enhance the exchange of opinions among peers. Therefore, teachers can arrange seats around a circle according to the size of the space and the number of students so that students can participate in face-to-face discussions. This setup can improve their attention and promote practical discussion. However, when conducting group discussions, attention should be paid to reducing the interference from multimedia and working sheets; otherwise, students may still concentrate on completing their study sheets instead of engaging with the group.

There were some limits to the study's effect and inference. First of all, the experimental group consisted of just sixteen students, which could limit the inference and generalizability

of the experimental effect. In addition, the control group students did not participate in the study or major in the gifted resource classroom's affective development course. The study could not compare how the intervention affected the two groups of talented students—mathematically or linguistically gifted—and how different gifted kinds were affected. Secondly, the tutorials impacted the experimental effect on integrative activities that both groups' students received in the regular lectures. Thirdly, students' ability to reflect on and transform themselves was hampered by the lack of sufficient time during the semester for discussion and sharing in the classes to adequately express their ideas and emotions to one another. Fourthly, because the affective development curriculum was an elective course, gifted students were strongly motivated to major in the course and might weigh outcomes and inflate the effect.Thus, the study also analyzed the worksheets to present students' viewpoints. Lastly, some scales employed originated from masters' theses, which could affect the reliability and efficacy of the scales.

The researcher suggests that interviews with a few students could be conducted randomly in addition to quantitative assessment and qualitative data analysis. Parents and tutors can also be involved in conversations so that they understand the pupils' typical emotional management circumstances. Confirming the intervention program's natural effect could result in a clearer understanding of how pupils typically apply new tactics. Concerning the inconsistent number of research participants in the pre–post-tests, the researcher could encourage gifted students to complete their post-tests online.

**Funding:** This research received no external funding.

**Institutional Review Board Statement:** The study originated from the author's doctoral thesis, and the research process followed the guidance of the research ethics. Ethical review and approval were waived for this study due to conducting the research without needing to add to ethical approval at that time.

**Informed Consent Statement:** Informed consent was obtained from all subjects involved in the study.

**Data Availability Statement:** The original contributions presented in the study are included in the article, further inquiries can be directed to the author.

**Acknowledgments:** Thanks to all participants in the study.

**Conflicts of Interest:** The author declares no conflicts of interest.

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
