# Peer review of "The Pilot Study on the Effect of a Compassion-Based Program on Gifted Junior High School Students’ Emotional Styles, Self-Compassion, Empathy, and Well-Being"

_education, doi:10.3390/educsci14101058_

Round 1
Reviewer 1 Report
Comments and Suggestions for Authors
The text is interesting from a scientific point of view, but it requires highlighting the research problems. In the presented methodological assumptions, they are not very visible. After correction, I recommend the material for publication.
Author Response
Thanks for your suggestion. I have added the description to adopt the second hypothesis in the first paragraph of the introduction to highlight the research problem. (lines 35 to 38)
Reviewer 2 Report
Comments and Suggestions for Authors
Summary
The manuscript titled ‘Study on the effect of a compassion-based program on gifted junior high school students' emotional styles, self-compassion, empathy, and well-being’ reports development and assessment of a compassion-based educational program for enhancing gifted junior high student’s positive psychological resources in Taiwan. The study is necessary since evidence suggests that these students are at risk of negative psychological impact from various factors and that without adequate coping resources their academic performance suffers. The manuscript presents the rationale, development and assessment of the program by means of quantitative analyses of pre- post-intervention with control group comparison, augmented by qualitative program feedback. For the students receiving the intervention, elements of self-compassion, empathy, and well-being were enhanced and participants reported many benefits. Based on the results and the authors’ experiences recommendations are made for implementing a similar program.
Specific constructive comments per section for the author’s consideration for manuscript revision and future submission are given below.
Title
Study on the effect of a compassion-based program on gifted junior high school students' emotional styles, self-compassion, empathy, and well-being
Consider reframing as a pilot-study (see later points)
Abstract
Number of participants is misleading, since the final numbers are lower than initially recruited. The specific elements improved by the program should be listed.
Introduction
Pg 2 lines 59 and 72 format of reference incorrect.
Pg 2 line 89 acronym SEL used without expansion in first instance.
The rationale is apparent but requires strengthening. Localisation / application to population being sampled requires support. Compassion section is introduced without an adequate feed-in.
In general, it is unclear how the theories and empirical evidence cited are integrated to justify a requirement for intervention and inform the education program design. Although the literature cited forms a sound evidence base, the introduction does not provide a coherent narrative of how it suggests the necessity for the present study, nor is there sufficient differentiation between reporting of elements of existing education programs and the design of the program implemented as an intervention in this study. Aims and hypotheses are not clearly stated.
Methods
The study quantitative design is sound to determine between-group differences whilst implementing suitable control for the confound of pre-test between-group difference performance.
The sample is not justified with a priori power analysis or from relevant literature. Although some information is provided regarding sample selection and attrition these are insufficiently attended to in the analysis and limitations section within the Discussion. For instance, self-selection could have weighted outcomes and inflated the effect. Whilst this is difficult to control for in such a scenario, mention of it should be made in the limitations section within the Discussion.
Pg 5 line 187 self-compassion scale does not have a subscale of ‘common humiliation’. As stated elsewhere in the manuscript, it is ‘common humanity.’
Several of the scales employed are not peer-reviewed, originating from masters theses. Whilst the journal does not prohibit citation of theses, they are not adequate to provide acceptable psychometric data. Although it is understandable that the authors sourced locally developed measures these are of restricted validity.
The intervention program is summarised, though more detail is required to enable adequate assessment of fitness or to enable replication. For instance, for how many hours per week did the sessions run?
Mention of qualitative analysis is inadequate to enable insight into to the method. Justification of its inclusion needs to be supported in the introduction. It should be a strength of the study that it is mixed methods if these are presented with parity.
Results
Report of the parametric tests to qualify the analysis are helpful and go some way towards reassuring the reviewer that findings may be treated with some level of confidence, noting the caution due to low sample size.
With this caveat in mind, very large effect sizes should be treated with due caution, though would suggest low sample size may be sufficient.
Discussion
The discussion integrates qualitative data to supplement the quantitative outcomes, though the qualitative data does not appear to have undergone any structured analysis, whether that be frequency based or interpretive.
Findings are contextualised with some relevant literature, though more recent evidence ought to be included.
Very numerous citation formatting errors, with (date) parentheses present.
Comments on the Quality of English LanguageIn general, the article requires extensive English language editing to correct mistaken words and improve grammar. Some sections are highly ambiguous and difficult to interpret. This reviewer is privileged to have English as a first language and has no command of any languages of Taiwan, so I applaud the successful translation that forms the bulk of the manuscript. However, there are sufficient issues to warrant rejection on the basis that editing is required.
Reviewer 3 Report
Comments and Suggestions for Authors
The topic of the article, ‘The effect of compassion-based programmes on emotional styles, self-compassion, empathy and well-being of gifted middle school students’, is a very current and important issue in the field of educational sciences. In particular, studies on gifted students are valuable because they offer innovative approaches to the social and emotional needs of these students.
The study was conducted with a quasi-experimental design including experimental and control groups. This shows that the article is based on a solid methodological foundation. However, the small sample size (17 experimental groups and 13 control groups) may limit generalisability. Explanations should be made about this situation.
The reliability and validity studies of the scales used in the study were well reported. This shows that the data collection tools used are academically acceptable. It is seen that detailed information on the cultural adaptation of the scales is not given. Explanatory information on this issue should be provided.
In the article, it is seen that a comprehensive literature review was conducted. Many references to gifted students' social and emotional development were taken as a reference. However, it can be evaluated that some of the references could be more up-to-date.
The findings demonstrated the differences between the experimental and control groups and it was emphasised that the results were statistically significant. However, discussing the findings in a broader context and comparing them with the literature is not sufficient. In this regard, the discussion section should be written in more detail.
Comments on the Quality of English LanguageThe article is written in accordance with academic writing rules. The language is fluent and professional. However, the complexity of sentence structures in some places may make it difficult to understand. These should be corrected.
Reviewer 4 Report
Comments and Suggestions for Authors
Though the area was interesting it was clear you identified gifted students. and this needs to be reworked on for your audience.
Your qualitative results should have been part of the actual results presentation to arrive at your findings for discussion

Comments on the Quality of English LanguageI suggest the authors should use proofreading services to improve the quality of the language.
Round 2
Reviewer 2 Report
Comments and Suggestions for Authors
Thank you for responding to the points raised following initial review. Having carefully read the revised manuscript and cover letter I am satisfied that the majority of concerns raised have successfully been attended to.
I maintain the SEL remains in need of expansion at first mention since SEAL and SEL are different, otherwise replace SEL with SEAL.
I would further suggest that the scale psychometric values must report the reliability for the present study, since values reported in the original scale development studies only pertain to the participants in those studies.
However, the decision to reject has to be made in light of inadequate ethics statement, as raised to the editor at the first round of review.
Comments on the Quality of English LanguageThe overall English Language of the revised manuscript is improved to the point that editorial publication team would be able to finalise without need for extensive revision.
Author Response
Comments 1: I maintain the SEL remains in need of expansion at first mention since SEAL and SEL are different, otherwise replace SEL with SEAL.
Response 1: I replaced SEL as SEAL in lines 105 and 107.
Comments 2: I would further suggest that the scale psychometric values must report the reliability for the present study, since values reported in the original scale development studies only pertain to the participants in those studies.
Response 2: The reliability and validity of the four scales have mentioned in research instruments.
Comments 3: However, the decision to reject has to be made in light of inadequate ethics statement, as raised to the editor at the first round of review.
Response 2: When I conducting the research, ethical review and approval were not requested from the Institutional Review Board Statement. So, I didn’t get the provident of the ethical review and approval. However, the research process was followed the guidance of the research ethics, and informed consent was obtained from all subjects involved in the study. In this way, I have revised statement of the Institutional Review Board Statement, as "The whole research process was followed the guidance of the research ethics."
Thanks for all the suggestions.
Reviewer 4 Report
Comments and Suggestions for Authors
Your manuscript has currently improved. However, give a second look to the integration of the qualitative results in the discussion of the findings. If possible, separate the qualitative results and merge the qualitative and quantitative findings in the discussion.

Comments on the Quality of English LanguageThe language has improved but there is always a need for improvement.
Author Response
Comment 1: The style of referencing raised in the previous review has not been address in Line 87
Response 1: The researcher deleted the front of the sentence, as “Kuo et al. (2020) found that j” in Line 87, and replace don’t as didn’t in Line 80.
Comment 2: Large body of empirical evidence with only one citation?
Response 2: The researcher added the references [41,42,43,44,46] in Line 104.
Comment 3: Check Pro-test in Table 2.
Response 3: The researcher revised as post-test in Table 2.
Comment 4: Why is your qualitative data now presented in this section? You have not addressed this concern raised in the previous review "Extract the qualitative data and present it with your quantitative results"
Response 4: The researcher has organized the qualitative data, and presented as the paragraph of "3.4. The feedback from students’ worksheets" before the discussion.
Thanks for all the suggestions.